# Detection of Elastic Deformation in Metal Materials in Infrared Spectral Range

**DOI:** 10.3390/ma14185359

**Published:** 2021-09-17

**Authors:** Milan Sapieta, Vladimír Dekýš, Ondrej Štalmach, Alžbeta Sapietová, Martin Svoboda

**Affiliations:** 1Faculty of Mechanical Engineering, University of Zilina, Unierzitna 1, 010 26 Zilina, Slovakia; vladimir.dekys@fstroj.uniza.sk (V.D.); ondrej.stalmach@fstroj.uniza.sk (O.Š.); alzbeta.sapietova@fstroj.uniza.sk (A.S.); 2Faculty of Mechanical Engineering, Jan Evangelista Purkyně University, 400 96 Ústí nad Labem, Czech Republic; svoboda@fvtm.ujep.cz

**Keywords:** infrared, thermoelastic, lock-in, mechanical load, experimental measurement

## Abstract

The aim of this work is to verify the presence of deformation in the metal specimen from the material AISI 316L by means of lock-in thermography. The specimen was cyclically loaded by the three-point bending in the fatigue testing machine. A response of the specimen to such excitation can be detected in the infrared spectrum and to determine temperature changes during a loading cycle. By means of the lock-in method, an increased signal to noise ratio (radiation energy detected by an infrared camera) was achieved. Besides, the temperature changes were determined on the basis of amplitudes of radiant energy changes detected by the camera. The temperature change (all radiant energy) corresponds with the first invariant of the tensor of deformation and, after a calculation and regarding the material parameters, also the invariant of the stress tensor. The proportionality between the signal from the camera and the first deformation invariant is achieved if the specimen load is an adiabatic event. This process is achieved by choosing a sufficiently high load frequency. In case of a presence origin of plastic deformations, there takes place only part of radiant energy. When we accept the hypothesis of a presence of just elastic deformations and plastic deformation is also present in the monitored process, then the evaluated thermograms based on the assumption of the presence of elastic deformation present anomalies in a distribution of the determined tensor invariant of deformations. These anomalies are caused by a presence of plastic deformations. Based on the anomalies, plastic deformation can be detected and subsequently analyzed. For the tested specimen and the applied load, the calculation of stress tensor was performed. It confirmed a congruence of results obtained by the analysis of the physical process in the infrared spectrum of the mid-wave infrared camera.

## 1. Introduction

Thanks to the greater availability and falling prices of infrared cameras, thermography has evolved in recent years from a rarely used technique to an increasingly popular method of research [1,2,3,4]. Military research has largely triggered the technical development of infrared cameras [5]. With the end of the Cold War, highly sensitive infrared technology became less and less subject to military restrictions. Today, it is a stable part of the civilian market [6,7,8].

Thermoelastic stress analysis has been used by engineers and scientists for more than 50 years to solve practical problems. It works on the principle of sensing the energy released during loading in the elastic region. When the load is removed, the body returns to its original position (elasticity) and its original temperature (thermoelasticity) [9].

Pulse thermography was invented for non-destructive testing of materials and components. Heat is generated by a pulse and is able to penetrate immediately below the surface of all objects, whether more or less, depending on the thermal capacity and thermal resistance of the material. The advantage of pulse thermography is that information on different depths can be obtained from a number of thermograms within a few seconds after the specimen is excited by the pulse [9,10]. In order for the induced thermal modulations generated by the input pulse to be measurable, they must exceed the noise level of the camera. This requires a high excitation pulse energy.

As an additional technique, lock-in thermography is offered, which is also known as the temperature wave imaging technique, as it can be described by the theory of oscillating temperature waves [11,12,13,14]. The excitation of heat in the body occurs periodically with a certain frequency [15]. The advantage of lock-in thermography is that with the registered image processing method used, it can significantly increase its sensitivity compared to the nominal sensitivity of the infrared camera used [16,17,18,19,20,21].

## 2. Thermoelastic Stress Analysis

Thermoelastic stress analysis (TSA) is an experimental contactless method based on measuring the infrared radiation emitted from the component surface exposed to dynamically linear elastic strain (deformation). Kelvin was the first scientist to study the thermoelastic effect, and the basic equations to describe the thermoelastic [2] effect were formulated by Darken and Curry [22].

The general form of the heat conduction equation for elastic body is derived from the energy conservation equation, and it can be written as follows [19]:(1)$${\rho c}_{\varepsilon }\frac{\mathrm{dT}}{\mathrm{dt}}-\frac{\partial }{\partial x_{j}}\left(k\frac{\partial T}{\partial x_{j}}\right)=\rho r+\sigma_{\mathrm{ij}}\frac{\partial \varepsilon_{\mathrm{ij}}}{\partial t}--\rho \frac{\partial \Psi }{\partial V_{k}}\frac{{\mathrm{dV}}_{k}}{\mathrm{dt}}+\rho T\frac{\partial^{2}\Psi }{\partial T\partial V_{k}}\frac{{\mathrm{dV}}_{k}}{\mathrm{dt}}.$$

Equation (1) uses the Einstein summation convention, ρ is the density, $c_{\varepsilon }$ is the specific heat capacity at constant deformation, T is the absolute temperature, k is the thermal conductivity tensor, $\sigma_{\mathrm{ij}}$ and $\varepsilon_{\mathrm{ij}}$ are the tensors of stress and strain, $\varepsilon_{\mathrm{ij}}$ is the inner heat source per unit of volume and Ψ is Helmholtz free energy that is dependent on k and independent of the internal state variables $V_{k}$.

Provided that the material elastic properties and constant material coefficients are temperature independent, then it can be derived a three-dimensional heat conduction equation written as follows [20]:(2)$${\rho C}_{\varepsilon }\dot{T}-{k\Delta }^{2}T=T_{0}\left(-\frac{E\alpha }{1-2\upsilon }\right){\dot{\varepsilon }}_{I}^{e}+\alpha_{p}\sigma_{\mathrm{ij}}{\dot{\varepsilon }}_{\mathrm{ij}}^{p}.$$

The equation for heat conduction will include the creation of the thermoelastic and the thermoplastic heat. Parameter $\varepsilon_{I}^{e}$ denotes the first invariant of elastic deformation tensor, $\varepsilon_{\mathrm{ij}}^{p}$ is the plastic part of the deformation tensor, α is the coefficient of thermal expansion, E Young’s modulus, υ Poisson’s ratio, and $T_{0}$ is the initial temperature [3]. Non-dimensional coefficient $\alpha_{p}$ is the ratio of the total plastic work to plastic work, which is converted to heat. The value $\alpha_{p}≅1$ because of only a small part of the plastic work (obtained energy at cold forming) is used to change the inner properties of the material [4]. Therefore, the thermoplastic area will be neglected and relationship will be formulated only for thermo-elastic area, which has the following form:(3)$$\Delta T=\frac{{\alpha T}_{0}}{{\rho c}_{p}}\Delta \sigma_{\mathrm{ii}}$$
where $c_{p}$ is the specific heat capacity under constant pressure and its relationship with $c_{\varepsilon }$ is as follows:(4)$$c_{\varepsilon }=c_{p}-\frac{2E\alpha^{2}T}{\rho \left(1-v\right)}.$$

## 3. Lock-In Method

A prerequisite for the use of this technology is that the excitation signal may periodically pulsate or in any other way modulated by certain dominant amplitude frequencies, called “lock-in frequency” $f_{\mathrm{lock}-\mathrm{in}}$. Assuming that the response to this excitation has the same frequency as the excitation or modulated excitation signal, ei, the response frequency coincides with the lock-in frequency, and we can successfully use the lock-in method to process the response signal for such excitation [8].

Lock-in thermography is a type of active lock-in method. It means that heat to excite the specimen is generated periodically and correlation using the lock-in process is applied to all heat signatures of every single pixel on the image of the observed object.

Digital lock-in process correlation generally consists of averaging the results of the measured values $F_{k}$ and system of weighting factors $K_{k}$ to the total number of measured values *M* [8]:(5)$$S=\frac{1}{M}\sum_{k=1}^{M}F_{k}K_{k}.$$
where S is the output signal.

If the excitation is harmonic then the most advantageous correlation function is also harmonic (sine, cosine) function. This kind of lock-in correlation is called sin/cos or narrowband correlation. It can be realized either by narrowing the bandwidth of the detected signal or using the values of harmonic functions for $K_{k}$ in Equation (5).

The main advantage of the sin/cos correlation is that it enables the user to take into consideration the phase of the signal after the measurement (off-line) when it is used with two-channel correlation. Two-channel correlation means that there were used two types of weighting factors, one approximates a sin-function and the second approximates cos-function. Correlation is transferred twice in parallel with both types of weighting factors [9]:(6)$$K^{0}\left(t\right)=2\sin\left(2\pi f_{lock-in}t\right),$$
(7)$$K^{\pi /2}\left(t\right)=2\cos\left(2\pi f_{lock-in}t\right).$$

Then, the first channel measures the component in-phase with the sin-function, and the other channel measures the component in-phase with the cos-function, which is π/2 phase-shifted to the sin-function [3].

If Equations (6) and (7) are inserted into Equation (5), then the result of the two correlations over the complete number periods is [8]:(8)$$S^{0}=\frac{1}{n}\sum_{i=1}^{n}F_{i}K_{i}^{0},$$
(9)$$S^{\frac{\pi }{2}}=\frac{1}{n}\sum_{i=1}^{n}F_{i}K_{i}^{\frac{\pi }{2}}.$$
where $\;S^{0}$ is called the in-phase signal and $S^{\frac{\pi }{2}}$ is usually called the quadrature signal. Both signals may be either positive or negative [4].

## 4. Setup for Stress Analysis of Beams Made of Stainless Steel

The measurement that was performed in this work was the measurement of the three-point bending of stainless steel beams. The specimen had a length of 50 mm and a square cross section measuring 10 × 10 mm. AISI 316L material was chosen for this measurement. A fatigue test machine was used as an excitation source, the lock-in frequency corresponded to the frequency of the sinusoidal compressive load, the response was detected by an infrared camera. The thermoelastic analysis according to equation (3) was also performed for the specimen, which was subsequently compared with the FEM analysis performed in the ANSYS program. For homogenization, the specimen surface was treated by emission spray. Although the spray is intended for long-wave cameras, it was sufficient for a partial correction of emissivity. To eliminate radiation reflections from the environment, the specimen storage area in the loading machine was lined with polystyrene boards. This procedure has proved to be more than sufficient in previous measurements of this type, so it has been used in this measurement and will most likely be used in other measurements if circumstances allow.

The specimen was placed on two supports with the loading force acting from above on the middle between the supports (Figure 1). The supports were placed at a distance of 29 mm from each other.

The maximum resolution of the IR camera FLIR SC7200 320 × 256 pixels and the corresponding native frame rate 383 Hz was selected. The loading frequency was 105 Hz. There was a slight subsampling, as previous measurements showed that this type of subsampling did not affect the quality of the results. Therefore, it was chosen as the most appropriate with respect to the resolution ratio and the sampling frequency.

The input material properties required for thermoelastic stress analysis of specimens made of AISI 316L material loaded by three-point bending were: $\propto =16.5\times \left[K^{-1}\right]$, $\rho =8000\;\left[\mathrm{kg}\cdot m^{-3}\right]$, $C_{p}=485\;\left[J\cdot {\mathrm{kg}}^{-1}K^{-1}\right]$, $K=4.25\times 10^{-6}\left[{\mathrm{MPa}}^{-1}\right]$.

In the experiment, five load cases were selected, due to better monitoring of increasing stress with increasing load force. The specimen was loaded cyclically on a ZWICK testing machine. The load was gradual, from the smallest to the greatest force. The sizes of loading forces are given in Table 1.

Static force is component of force that was specimen preloaded and dynamic force is component of force that cyclically loads the specimen. The static component of the loading force was always chosen to be larger in order to avoid unloading of the beam and possible displacement on the supports.

## 5. Verification of Results Using FEM Analysis

In the next part, the results of thermoelastic analysis were verified using FEM simulation. To achieve the most reliable results, the simulation was solved as a contact task. The solution uses plane stress and half symmetry. This means that only half of the specimen was modeled and boundary conditions for symmetry were taken in the middle. The results are drawn as a whole model (Figure 2).

Two rigid members were modeled, the first simulating one support and the second a loading finger. The support was placed under the specimen at a distance of 14.5 mm from the center. The second member simulating the loading finger was positioned to push on the center of the specimen. Also, only half of the load finger was modeled, a point was selected in the middle of the quarter of the circle, which was chosen as the control point, and boundary conditions such as symmetry and load were prescribed for it. The dimensions of both rigid members were accurately measured by the radii of the loading finger and the support used in the experiment.

The material properties used in this simulation were Young’s modulus of elasticity 200 GPa and Poisson’s number 0.3.

In this work, the results are compared for the material AISI 316L for the selected load 0.5, 2, and 4 kN. Loads are selected for minimum, medium, and maximum value for a given material type. 

After performing the analysis, the distribution of principal stresses was achieved (Figure 2). The given stress distribution is for loading with a force of 4 kN. It is possible to observe the maximum value of the stress on the underside of the specimen below the point of loading marked as MX. The minimum stress value is marked MN on the upper part of the specimen just below the place of loading. At the loading points on the upper side of the specimen and at the attachment points on the supports, plastic areas were created during the numerical simulation. These plastic areas are marked in gray (Figure 2). The formation of areas with a higher stress value at the places of loading and attachment is characteristic for all loads.

The stress field distribution in the numerical simulation and the stress field distribution in the experiment are compared on a specimen with a load of 0.5 kN (Figure 3). In both cases, the positive stress values are located on the underside of the specimen marked in red. Negative stress values are located at the top of the specimen marked in blue. It is possible to observe the symmetry according to the vertical plane.

An interesting feature of this comparison is the occurrence of the so-called blue dot in the experimental measurement (Figure 3). In numerical simulations, it is possible to observe similar points above the storage locations in all simulations. It is currently unclear how this point was captured in experimental measurements. In the case of repeated measurements, it would be appropriate to control possible reflections to a greater extent and then—if necessary—to eliminate or minimize them.

In the case of negative stresses on the upper side of the specimen, it is possible to observe a slight rise in stress below the point of loading visible in pale blue. This manifestation is characteristic of both experimental measurement and numerical simulation. The maximum value of the stress in the experiment was 20.49 MPa and in the numerical simulation 19.75 MPa, which represents a difference of 3.6%. At the minimum value, this difference was already significant, mainly due to the formation of high values of stress below the load site in the numerical simulation. For all types of specimens and loads, similar differences in maximum and minimum stress values can be mentioned.

Further comparisons were made in the form of graphs Figure 4, Figure 5 and Figure 6 for selected loads of 0.5, 2, and 4 kN.

The course of stress on the line was evaluated depending on the position on the specimen and the position on the geometric model of the simulation, respectively. The line for which the stress curves were evaluated passes through the center of the specimen in the vertical direction and can be seen in Figure 2 and Figure 3 in black.

The graph for the first load of 0.5 kN is characterized by two curves. Blue for the stress course on the experimental measurement line and red for the stress course on the same line, but on the numerical simulation model (Figure 4). Both curves start at approximately the same point around 20 MPa. They have the same descending character after a position of about 6.5 mm. There they begin to separate (Figure 4). At approximately the eighth millimeter, there is a sharp drop in the stress value in the numerical simulation, which represents a gray area for the distribution of the stress field below the load site (Figure 4).

On the graph for a load of 2 kN, it is possible to observe both curves starting from the same place, which represents the maximum values of stress (Figure 5). Both curves have the same decreasing character up to about 7.8 mm. Here, the curves are separated and the curve for the numerical simulation begins to decline sharply due to a stress drop at the load location.

The graph for the 4 kN load has a similar course as the two previous graphs. Again, a sharp drop in stress due to the formation of a plastic region in the numerical simulation can be observed in the 8th mm region (Figure 6).

The differences between the experiment and the computation in Figure 4, Figure 5 and Figure 6 is explained by the computation of high contact stresses in the vicinity of the applied force, but the used thermoelastic analysis in ALTAIR LI approaches this area as a classical elastic. The comparison of deviations between measurement and simulation is given in Table 2. The table compares only the values for the maximum stress due to the formation of contact stress in numerical simulations in the area of minimum stress.

## 6. Discussion

Thermoelastic stress analysis deals with the problem of obtaining stress fields on the surface of a given object using an IR camera. The measurement was performed in Altair LI software, which results in images of stress fields. In this measurement, the results of the experimental part were compared with the numerical solution. Numerical analysis was performed taking into account the contact where the specimen was modeled as a flexible body and the load finger and supports as rigid bodies. In comparison, it was found that the distribution of stress fields of the first stress invariant on the test model and the experimental specimen match. We want to use the plasticity violation of the elastic model in the detection and further analysis of the plastic area. In this procedure, we avoided determining the exact dependence between the detected energy and the deformation for the plastic area.

If plastic deformations occur, the relation (3), which applies to the elastic region, is broken, and then we want to detect and analyze the plastic region by a combination of experiment and FEM calculation.

## Figures and Tables

**Figure 1 materials-14-05359-f001:**
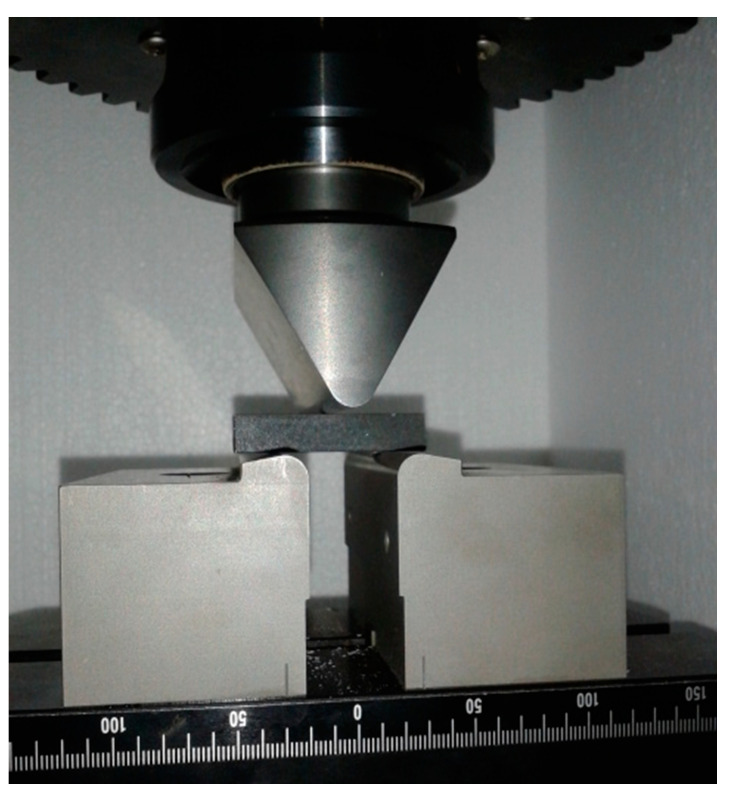
Specimen position under three-point bending loading.

**Figure 2 materials-14-05359-f002:**
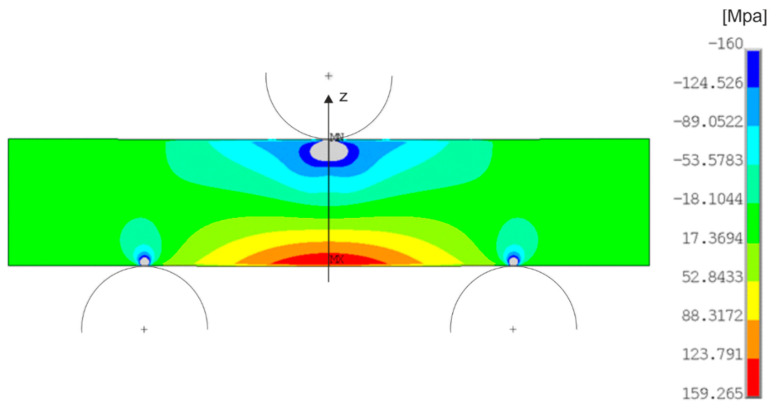
Distribution of principal stresses numerical analysis for material AISI 316L and load 4 kN.

**Figure 3 materials-14-05359-f003:**
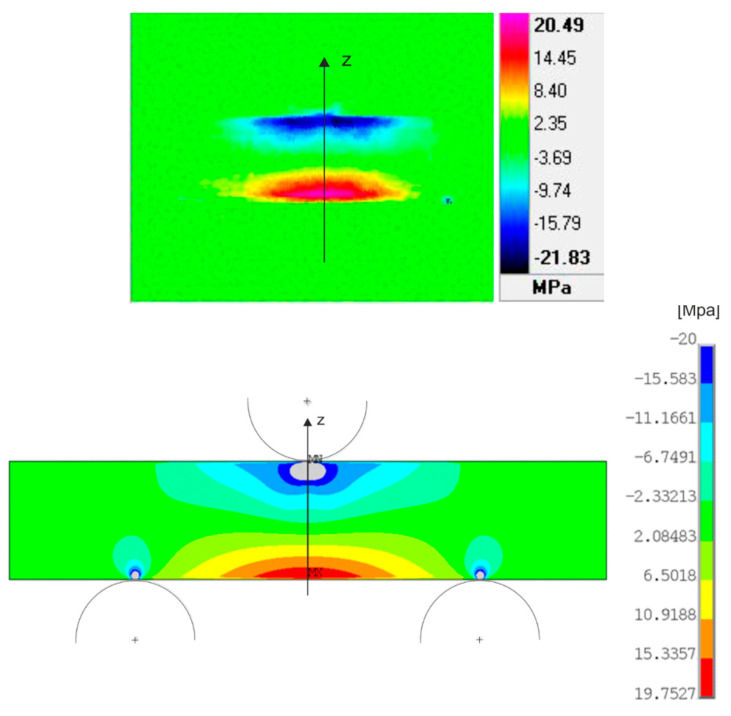
Comparison of the distribution of principal stresses from experimental measurements with the results of numerical analysis, material AISI 316L, and load 0.5 kN.

**Figure 4 materials-14-05359-f004:**
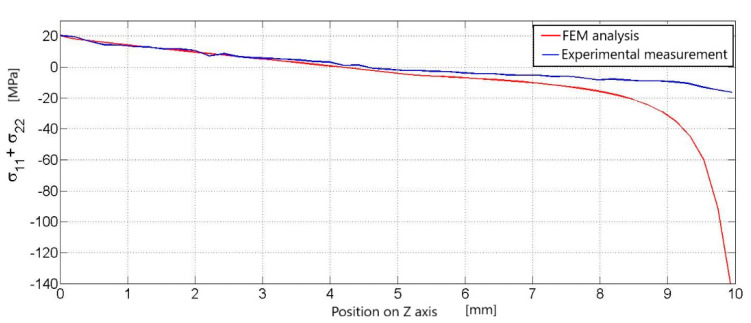
Comparison of the course of principal stresses in experimental measurements with the course of principal stresses in numerical analysis, waveforms according to the vertical black line (Figure 3), compared results for a load of 0.5 kN.

**Figure 5 materials-14-05359-f005:**
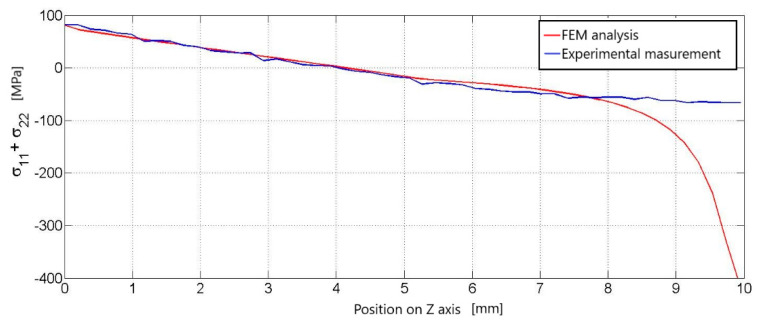
Comparison of the course of principal stresses in experimental measurements with the course of principal stresses in numerical analysis, courses along the black line (Figure 3), compared results for a load of 2 kN.

**Figure 6 materials-14-05359-f006:**
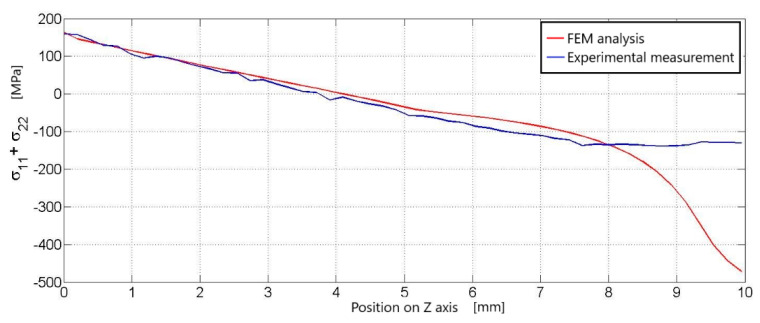
Comparison of the course of principal stresses in experimental measurements with the course of principal stresses in numerical analysis, courses along the black line (Figure 3), compared results for a load of 4 kN.

**Table 1 materials-14-05359-t001:** Selected load cases for three-point bending.

Load Case	1.	2.	3.	4.	5.
Static force component [kn]	1	2	3	4	5
Dynamic force component [kn]	0.5	1	2	3	4

**Table 2 materials-14-05359-t002:** Comparison of the deviation of the results of numerical simulation and experimental measurement.

Load [kN]		Experimental Measurement [MPa]	Numerical Simulation [MPa]	Deviation in %, [1]
0.5	Max.	20.49	20.29	0.97
2	Max.	82.07	81.14	1.13
4	Max.	159.01	163.61	2.81

## Data Availability

Not applicable.

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
