# Peer review of "Detection of Elastic Deformation in Metal Materials in Infrared Spectral Range"

_materials, 2021, doi:10.3390/ma14185359_

Round 1

Reviewer 1 Report

This paper describes the detection of elastic deformation (or evaluating stress fields) in an AISI 316L steel using infrared spectrum by means of a lock-in thermography. Both experiment results and FEM are provided.

Though the topic might be of interest to some readers, this paper is poorly written with its English and logic hard to follow. Also, the paper fails to demonstrate a clear research gap.

  1. The title “Detection of Deformation in Metal Materials in Infrared Spectral Range”, the author might want to change the “deformation” to “elastic deformation” to better reveal it’s content.
  2. In the Abstract, the first three sentences are ok, but then the authors presented many terminologies without clearly demonstrating the logic between them and I am confused after reading them,

“By means of the lock-in method was increased signal – to noise ratio and the temperature changes determined. The temperature change corresponds with the first invariant of the tensor of deformation and, after a calculation and regarding the material parameters, also the invariant of the stress tensor. In case of an adiabatic process that will be reached in the suitable frequency and by the load that activates only elastic deformations in the tested specimen, we speak of a temperature modification that is directly proportional to the invariant tensor of deformations. In case of an origin of plastic deformations, there takes place only partial energy radiation by which the specimen was excited. When we accept the hypothesis of a presence of just elastic deformations, then the evaluated thermo-grams present anomalies in a distribution of the determined tensor invariant of deformations and these anomalies are caused by a presence of plastic deformations. For the tested specimen and the applied load, the calculation of stress tensor was performed. It confirmed a congruence of results obtained by the analysis of the physical process in the infrared spectrum of the mid-wave infrared camera.”

The authors need to rewrite these sentences to better demonstrate the importance and the most significant conclusions of the work and also keep the consistency of their expression.

  1. Moving to 1. Introduction,

In the third paragraph of the, the authors suddenly jump to the advantage of “pulse temography” (is temography a typo?), what are the connection with the first two paragraphs? I have no clues…

Then again they jump to the “lock-in thermography”, though I understand it might be a method to tackle the disadvantage of the “pulse tomography”, no comparison with the “pulse temography” was mentioned by the authors…

The authors also need to clearly demonstrate the research gap (which is now lacking) in this part. I believe the application of lock-in thermography in this analysis is not something new, as shown in the paper’s references [16-22], what is new in this paper when the current work is placed in the literature? Is this method first used in thermoelastic analysis?

  1. In the 2. Thermoelastic stress analysis part,

In the first paragraph, they authors mentioned the work of Darken and Curry, however, no reference was provided.

  1. In the 4. stress analysis of beams made of stainless steel

The font in the last paragraph is not consistent with other parts.

 In Figure 1, it looks like that the specimen is not placed symmetrically with respect to the loading force and the two supporting parts. Why?

Please explain the static force and dynamic force components in Table 1.

No results of the stress analysis were provided in this part (which is not in agreement with its part title “stress analysis of beams made of stainless steel”), though I understand that these results were kind of provided in comparison to the FEM simulation result. This is strange.

  1. In the part 3.1. Verification of results using FEM analysis,

The number of this part is not correct.

In the third paragraph “The material properties used in this simulation were Young's modulus of elasticity 200 x 10-3”? Do the author mean the Young’s modulus of the material is 200x10-3? If so, the authors need to correct it…

In Figures 2-6, no magnitude of the legend is provided! The z-axis needs to be defined.

  1. Discussion part,

”I assume that other values are caused by averaging, resp. filter used in the measurement.” A confusing sentence…..

Therefore, I cannot recommend its publication.

Author Response

We thank the reviewer for their supportive comments and the time they spend reviewing our submission. We very much appreciate their efforts to improve the submission. Next, we explain the changes we made to the manuscript based on the received comments. All the changes in the revised submission are marked with red.

This paper describes the detection of elastic deformation (or evaluating stress fields) in an AISI 316L steel using infrared spectrum by means of a lock-in thermography. Both experiment results and FEM are provided.

Though the topic might be of interest to some readers, this paper is poorly written with its English and logic hard to follow. Also, the paper fails to demonstrate a clear research gap.

  1. The title “Detection of Deformation in Metal Materials in Infrared Spectral Range”, the author might want to change the “deformation” to “elastic deformation” to better reveal it’s content.

Response 1: The title changed to “Detection of Elastic Deformation in Metal Materials in Infrared Spectral Range”

  1. In the Abstract, the first three sentences are ok, but then the authors presented many terminologies without clearly demonstrating the logic between them and I am confused after reading them,

“By means of the lock-in method was increased signal – to noise ratio and the temperature changes determined. The temperature change corresponds with the first invariant of the tensor of deformation and, after a calculation and regarding the material parameters, also the invariant of the stress tensor. In case of an adiabatic process that will be reached in the suitable frequency and by the load that activates only elastic deformations in the tested specimen, we speak of a temperature modification that is directly proportional to the invariant tensor of deformations. In case of an origin of plastic deformations, there takes place only partial energy radiation by which the specimen was excited. When we accept the hypothesis of a presence of just elastic deformations, then the evaluated thermo-grams present anomalies in a distribution of the determined tensor invariant of deformations and these anomalies are caused by a presence of plastic deformations. For the tested specimen and the applied load, the calculation of stress tensor was performed. It confirmed a congruence of results obtained by the analysis of the physical process in the infrared spectrum of the mid-wave infrared camera.”

The authors need to rewrite these sentences to better demonstrate the importance and the most significant conclusions of the work and also keep the consistency of their expression.

Response 2: The abstract is rewritten for a better understanding of the topic of the article.

  1. Moving to 1. Introduction,

In the third paragraph of the, the authors suddenly jump to the advantage of “pulse temography” (is temography a typo?), what are the connection with the first two paragraphs? I have no clues…

Then again they jump to the “lock-in thermography”, though I understand it might be a method to tackle the disadvantage of the “pulse tomography”, no comparison with the “pulse temography” was mentioned by the authors…

The authors also need to clearly demonstrate the research gap (which is now lacking) in this part. I believe the application of lock-in thermography in this analysis is not something new, as shown in the paper’s references [16-22], what is new in this paper when the current work is placed in the literature? Is this method first used in thermoelastic analysis?

Response 3: Yes temography is a typo.

I put it wrong, I added a few sentences there to make it easier to understand.

Lock-in thermography is a method for increasing the sensitivity of an infrared camera. Lock-in can also be used for pulse temography. 

New in this paper is used lock-with thermoelastic analysis and use infrared camera with max. resolution 320x256 pixels. By using the lock-in method we get a much higher sensitivity of the infrared camera and thus more accurate results.

  1. In the 2. Thermoelastic stress analysis part,

In the first paragraph, they authors mentioned the work of Darken and Curry, however, no reference was provided.

Response 4: Reference of the work of Darken and Curry added to article. 

  1. In the 4. stress analysis of beams made of stainless steel

The font in the last paragraph is not consistent with other parts.

 In Figure 1, it looks like that the specimen is not placed symmetrically with respect to the loading force and the two supporting parts. Why?

Please explain the static force and dynamic force components in Table 1.

No results of the stress analysis were provided in this part (which is not in agreement with its part title “stress analysis of beams made of stainless steel”), though I understand that these results were kind of provided in comparison to the FEM simulation result. This is strange.

Response 5: The font is corrected now.

The specimen is placed symmetrically in Figure 1. The photo is taken at an angle it can give the impression that the specimen is not symmetrically placed while it is. The photo perpendicular to the specimen could not be taken because an infrared camera was placed there.

Static force is component of force which was specimen preloaded and dynamic force is component of force which cyclically loads the specimen.

Part title was change to “Setup for stress analysis of beams made of stainless steel

  1. In the part 3.1. Verification of results using FEM analysis,

The number of this part is not correct.

In the third paragraph “The material properties used in this simulation were Young's modulus of elasticity 200 x 10-3”? Do the author mean the Young’s modulus of the material is 200x10-3? If so, the authors need to correct it…

In Figures 2-6, no magnitude of the legend is provided! The z-axis needs to be defined.

Response 6: The number of this part was changed.

Young's modulus of material is written correct now.

Legend is provided now and also the z-axis is defined.

  1. Discussion part,

”I assume that other values are caused by averaging, resp. filter used in the measurement.” A confusing sentence…..

Response 7: The sentence was replaced.

Reviewer 2 Report

The present manuscript demonstrates the possibility to measure stress in a metallic three-point bending specimen by means of lock-in themography. The results nicely describe the technique, and demonstrate the applicability for a material relevant for engineering. The results are verified using FEM showing good correlation.

The manuscript is well written but the figures can be improved. Specifically, axis and color scales have no units. In addition, the figure caption of Figure 2 is wrong.

Author Response

We thank the reviewer for their supportive comments and the time they spend reviewing our submission. We very much appreciate their efforts to improve the submission. Next, we explain the changes we made to the manuscript based on the received comments. All the changes in the revised submission are marked with red.

The present manuscript demonstrates the possibility to measure stress in a metallic three-point bending specimen by means of lock-in themography. The results nicely describe the technique, and demonstrate the applicability for a material relevant for engineering. The results are verified using FEM showing good correlation.

The manuscript is well written but the figures can be improved. Specifically, axis and color scales have no units. In addition, the figure caption of Figure 2 is wrong.

Units in figures are provided now and also z-axis. Caption of Figure 2 is corrected.

Round 2

Reviewer 1 Report

No significant scientific issues. I will leave it to the editors to decide whether to ask the authors to revise the language and the format of the paper.

For example, this sentence in the introduction "By means of the lock-in method was increased signal – to noise ratio (radiation energy detected by infrared camera) and the temperature changes was determined on the base of amplitudes of radiant energy changes detected by the camera."

This sentence doesn't make sense to me. It should be something like, "By means of the lock-in method, an increased signal – to noise ratio (radiation energy detected by an infrared camera) was achieved. Besides, the temperature changes were determined on the basis of amplitudes of radiant energy changes detected by the camera."

In terms of format, for example, you can find the there are still two parts under the same part 4 listed below. I don't see that the authors have paid enough attention to this.

4. Setup for stress analysis of beams made of stainless steel 

4. Verification of results using FEM analysis 

Author Response

We thank the reviewer for their supportive comments and the time they spend reviewing our submission. We very much appreciate their efforts to improve the submission.

For example, this sentence in the introduction "By means of the lock-in method was increased signal – to noise ratio (radiation energy detected by infrared camera) and the temperature changes was determined on the base of amplitudes of radiant energy changes detected by the camera."

This sentence doesn't make sense to me. It should be something like, "By means of the lock-in method, an increased signal – to noise ratio (radiation energy detected by an infrared camera) was achieved. Besides, the temperature changes were determined on the basis of amplitudes of radiant energy changes detected by the camera."

Your sentence sounds better we replaced it in the article.

In terms of format, for example, you can find the there are still two parts under the same part 4 listed below. I don't see that the authors have paid enough attention to this.

4. Setup for stress analysis of beams made of stainless steel 

4. Verification of results using FEM analysis

We have already fixed it.